# A multi-objective mathematical model of a water management problem with environmental impacts: An application in an irrigation project

**G. M. Wali Ullah**[1☯]*, **Micah Nehring**[2☯]

**1** Department of Mathematics, University of Chittagong, Chattogram, Bangladesh, **2** School of Mechanical and Mining Engineering, University of Queensland, Brisbane, QLD, Australia

☯ These authors contributed equally to this work.
* wali@cu.ac.bd

**Data Availability Statement:** All relevant data are within the manuscript and its S1–S4 Tables files.

**Funding:** The author(s) received no specific funding for this work.

## Abstract

The study proposes applying an efficient but straightforward multi-objective constrained optimization model for optimal water allocation among irrigation and environmental sectors. The model has been implemented in the Muhuri Irrigation Project (MIP), Bangladesh, where the irrigation systems lead to unjustifiable use of groundwater. This study explores how water can be optimised to increase agricultural production and sustain the local environment in the MIP. Hence, the paper has two objectives—to maximise the net return and minimise the deficit in environmental flow. The study uses a Non-Dominating Sorting Genetic Algorithm, NSGA-II, to solve the research problem. Results indicate that crops more profitable to trade should be cultivated. Furthermore, the rainfall has more impact on the net return and environmental flow deficit than water inflow. The findings of this study can help plan irrigation water and cropland resources and be a reference for further studies.

## 1. Introduction

The scarcity of water is one of the significant issues in the agricultural sector in Bangladesh. Although Bangladesh is low-lying, riverine and featured by heavy rainfalls, the country suffers from seasonal water scarcity, especially during winter. The agriculture sector is the highest user of water in Bangladesh. This sector uses about 88% of total available water [1]. However, irrigated agriculture has begun in the 1960s. With the introduction of plentiful varieties of crops and the irrigation systems' modernisation, cultivation through irrigated water has become widespread [2].

Bangladesh is a low-lying country with an area of approximately $144, 170 \text{ km}^2$. From a climatic perspective, Bangladesh has four main seasons in a year: (i) from December to February is the dry winter season, (ii) from March to May is summer, the hot and humid season, (iii) from June to September is the rainy monsoon season, and (iv) from October and November is the autumn season. Summer in Bangladesh is very humid as winds blow from the southern

**Competing interests:** The authors have declared that no competing interests exist.

hemisphere, creating a lot of moisture in the atmosphere, eventually depositing heavy precipitation amounts. In contrast, winds from the northern hemisphere are arid and cold; these blow towards the warm southern oceans. Of the total rainfall in Bangladesh, about 71% occurs in the rainy season, 27% occurs in the summer and autumn, and 2% occurs in the winter [3]. Still, rainfall in summer and autumn is rare. That is why Bangladesh faces two extreme water-related events each year, namely flood and drought [4]. For cultivating required crops in periods of dry and unreliable rainfall, the country needs to increase water-use efficiency and water conservation.

Moreover, Bangladesh is a densely populated country. Its population is about 168 million. Approximately 37.2% of this population live in urban areas, and 62.8%, living in village areas. Bangladeshi villages are still agrarian. The villagers rely on agriculture and agricultural productivity to earn their livelihood and lead their life [1].

The paper engages with an agricultural project in Bangladesh, known as the Muhuri Irrigation Project (MIP), in light of the above background. This project is in Feni, a south-eastern district in Bangladesh, around the confluence of Feni, Muhuri and Kalidaskhali rivers in the coastal belt of the Bay of Bengal. The MIP consists of a closure dam and a 20-vent regulator. This project's construction began in 1978 and was completed in 1986 at the cost of $40 million. Building this project has been to provide irrigation facilities during the winter and regulate the inflow of saline water from the Bay of Bengal into the fresh river water. The project also helps farmers grow various kinds of crops in the dry season on the banks of the Muhuri River. It also functions as a large water vessel to produce many varieties of local fish [5].

However, the MIP tends to be less productive and beneficial than was initially planned. Its water cannot be used for the cultivation of crops optimally during dry periods. The lack of a proper water supply system, poor drainage, and unplanned cropping intensities appears to be some drawbacks in making the most out of the project. Nevertheless, this project built for the betterment of the agricultural community has never drawn any academic attention. No researchers have engaged with its drawbacks or potentials either inside or outside of Bangladesh. Therefore, the present paper finds MIP an exciting research area and identifies water allocation as a research problem. In this way, the article fills a research gap and finds a solution to the water allocation problem while maintaining a balance between water and natural life within the MIP. The Lewis and Randall [6] model is adopted and improved for this research project and uses a Non-Dominating Sorting Genetic Algorithm, NSGA-II, to solve the problem. This research uses a Multi-objective Optimisation Problem (MOP) in the agriculture sector in the MIP. Thus, the study locates at the intersection of mathematics and agriculture. Its findings can contribute to the optimal distribution and allocation of water to grow agricultural production in the Feni locality.

Multi-objective Optimisation Problem (MOP) has application in water management, agriculture, industry, engineering, economics, mining and many other fields where the problem involves simultaneously optimising several conflicting objectives. For example, in agriculture, the application of multi-objective optimisation models is well-accepted.

In recent decades, researchers from various parts of the world such as Australia, India, Iran, South Africa, China, Pakistan, and Saudi Arabia have developed their models or built their agricultural water allocation research on existing multi-object optimisation models [6–16]. Lewis and Randell [6] used multi-objective evolutionary computational techniques and Pareto optimisation concepts to solve different decision problems, including environmental flow in the agricultural system of the Irrigation area at Berembed weir on the Murrumbidgee River, Australia. Wardlaw and Bhaktikul [7] developed a Genetic Algorithm (GA) for solving multi-objective water scheduling problems in irrigation in the Indira Ghandi Nahal Pariyonaja (IGNP) irrigation system located in North-West India. A rotational basis operating system is

applied for optimising the water resources in the irrigation systems. Again, Xevi and Khan [8] used a multi-objective decision-making structure for solving water allocation problems with conflicting objectives in irrigation. The three conflicting objective functions of the model are minimising variable cost, maximising net return, and minimising total pumping requirements for supplementary groundwater [8]. The authors used a goal programming model with a weighted version where a single objective function is created by combining all three objective functions using different weights to solve the MOP. Ikudayisi et al. [10] presented a combined Pareto multi-objective differential evolution algorithm to optimise crop distribution and water allocation in the irrigation under inadequate water accessibility at the Vaal-Harts Irrigation Scheme (VIS) in South Africa. They used two conflicting objective functions: minimising total water allocation in the irrigation and maximise net benefit. Musa [14] applied a multi-objective model in Saudi Arabia for optimal water allocation in three sectors named domestic sector, agriculture sector, and industrial sector. A goal programming technique has been used to solve this problem. Marzban et al. [15] proposed an optimal cropping pattern of irrigation and rainfed crops by using multi-objective nonlinear programming to minimise environmental impact and maximise the revenue in Iran.

The present article builds on the Lewis and Randell model [6] to solve a multi-objective optimisation problem in water allocation in the Muhuri Irrigation Project, Bangladesh. It uses a Non-Dominating Sorting Genetic Algorithm, NSGA-II, to solve the nonlinear constraint problem to find the optimum result. This model was applied to data sourced from the literature and the Bangladesh Water Development Board (BWDB), Feni, Bangladesh. The main aims of the study are to maximise net return and minimise the deficit in environmental flow by adjusting irrigation water when seasonal water availability is limited.

The main contributions of this article are as follows.

i.  The Lewis and Randall [6] model is adopted and improved for this research project and applied in the Muhuri Irrigation Project (MIP), Bangladesh.

ii.  Considering the scenarios of different available water resources, the results can impact the agricultural production in the MIP area.

iii.  This method is very systematic and applied to different scopes, including water resources management. However, the most important thing is that the model can be used in other irrigation projects only by modifying the parameters according to the actual situation.

The remainder of this work is organised as follows: Section 2 presents the multi-objective optimisation problem, Section 3 explores the mathematical formulation, Section 4 contains the model solution and experimental format, Section 5 illustrates the results, and finally, Section 6 presents the conclusion of the study.

## 2. Multi-objective optimisation problem

Optimisation refers to maximising a system's desirable characteristics while minimising its undesirable properties [17]. Optimisation can be both single-objective and multi-objective. Still, the multi-objective optimisation model, which this research adopts, tends to be most suitable for solving real-world problems. These mainly involve several contradictory and conflicting objectives. Multi-objective Optimisation Problems (MOPs) indicate optimisation problems with more objective functions that have to be optimised systematically and simultaneously under a given feasible region. MOPs are essential for our real-life because they provide a model for the case in which we have to consider the trade-off of several conflicting objectives.

To optimise all objective functions simultaneously and find a unique solution in real-life problems is difficult. Let us consider the following MOP

$$\min f(x) \tag{1}$$

$$\text{s.t. } g_j(x) \leq 0, \ j = 1, 2, \ldots, m$$

where $f(x) := [f_1(x), \ldots, f_l(x)]$ stands for a vector of $l$ objective functions and $x \in \mathbb{R}^n$, where $f_i$: $\mathbb{R}^n \rightarrow \mathbb{R}$, $i = 1, \ldots, l$, and $g_j$: $\mathbb{R}^n \rightarrow \mathbb{R}$, $j = 1, \ldots, m$.

The solutions of (1) are called *Pareto points* [18] or *efficient points* [19] or *nondominated* solutions. A point $\bar{x} \in X$ is said to be an *efficient point* or *Pareto point* or *nondominated* solution for Problem (MOP) iff there is no $x \in X$, such that $f_i(x) \leq f_i(\bar{x})$, $\forall_i \in \{1, \ldots, l\}$, and $f_j(x) < f_j(\bar{x})$ For some $j \in \{1, \ldots, l\}$. The plot consisting of the images of these *Pareto points* in the performance (objective) region is called the *Pareto front*. When we cannot find any better solution in value without sacrificing some of the other objective values, the solution is called a *Pareto optimal* solution. From the mathematical perspective, all Pareto optimal solutions are equally acceptable as the MOP solution. Nevertheless, in the end, only one solution will be chosen out of the *Pareto optimal* set. The choice made to choose a desirable solution depends on a decision-maker. Someone who takes the position of the decision-maker knows the inner parts of the problem and can convey their preference relations between different solutions. However, options have to be given to the decision-maker first for them to decide.

## 3. List of symbols

| | |
|---|---|
| $X_c$ | Area of crop c to be planted in a hectare |
| $Env\_flow\_f(m)$ | Environmental flow for a month (m) |
| $TCI_c$ | Total crop income |
| $C_p$ | Groundwater pumping and delivery cost per unit volume |
| $P_m$ | Groundwater pumping volume in a month (m) |
| $Vcost_c$ | Variable cost (such as seeds, fertiliser, labour, and pesticides) per hectare excluding water cost for the crop (c) |
| $C_w$ | Water supply costs from dam per unite volume |
| $WREQ_{c,m}$ | Water requirement for the crop (c) in a month (m) |
| $C$ | Total number of types of crops to be planted |
| $M$ | Total number of months in the planning period |
| $Ter\_env\_flow\_f(m)$ | Target environmental flow for a month (m) |
| $k_{c,m}$ | Crop coefficient for the crop (c) in a month (m) |
| $ET_m$ | Evapotranspiration for a month (m) |
| $Rain_m$ | Rainfall for a month (m) measured by millimetres |
| $total\_Pump$ | Permissible pumping for the year in the irrigated areas |
| $T_{Area}$ | Total cropping area available |
| $minimum\_area$ | Minimum plantable areas |
| $Allocation(m)$ | Surface water amount which is accessible for irrigation in a month (m) |
| $Inflow(m)$ | Amount of surface (river) water available for a month (m) |

## 4. Mathematical formulation

In this section, we present a water management model introduced by Lewis and Randall [6]. A description of the mathematical expressions used to construct the two-objective optimisation

model is provided. Our goal is to formulate terms that measure the net return (NR), the shortage of irrigation water, and the environmental flow deficit (EFD).

In this article, we aim to find the planting areas per crop and corresponding optimal crop mix while maximising net return (NR) whilst minimising irrigation water and minimising deficit in environmental flow (EFD). The decision variables are $X_c$ and $Env\_flow\_f(m)$. The first objective of the model is to maximise net return (NR)

$$\max NR = \sum_{c=1}^{C} TCI_c\, X_c - C_p \sum_{m=1}^{M} P_m - \sum_{c=1}^{C} Vcost_c X_c$$

$$-C_w \sum_{m=1}^{M} \left( \left( \sum_{c=1}^{C} WREQ_{c,m} X_c \right) - P_m \right). \tag{2}$$

The first term of the objective function in Eq (2) is the total revenue and the second term is the expenditure related to the groundwater pumping and delivery cost. The third term is the expenditure, which comprises the variable cost such as fertiliser, pesticides, seeds, and other costs. Finally, the last term is related to the expenditure, including the cost of surface water supply accessible for irrigating crops in a month (m). The difference between the revenue and all expenditures gives the net return.

The second objective is to maintain enough downriver flows to sustain the environment. This objective is set to maintain a balance between water use and the life of nature in the MIP. Because if the focus is given only on irrigation but not on its environment, biodiversity will be hampered. Still, the objective focuses on how to sustain bio-diversity with minimum use of water.

$$\min EFD = \sum_{m=1}^{M} \max[(Ter\_env\_flowf(m) - Env\_flow\_f(m)), 0] \tag{3}$$

The only terms in the summation of Eq (3) included are only for those months where the environmental flow is less than the target; otherwise, zero is used instead. The environmental flow, $Env\_flow\_f(m)$ is the river's flow pattern necessary to sustain the ecosystem.

## Water requirement

The crop water requirements per month, $WREQ_{c,m}$, is the excess of evapotranspiration with the growth duration in months over rainfall,

$$WREQ_{c,m} = k_{c,m}\, ET_m - Rain_m. \tag{4}$$

## Problem constraints

There are several environmental and physical constraints imposed on the model, which are shown below.

The first constraint is the pumping water constraint,

$$\sum_{m=1}^{M} P_m \leq total\_Pump. \tag{5}$$

This constraint ensures that pumped groundwater does not exceed the allowable pumping for the year from the irrigation area.

The second constraint is the maximum area constraint,

$$\sum_{c=1}^{C} X_c \leq T_{Area}. \tag{6}$$

This constraint limits the total crop area planted to be equal to or less than the total area available.

The third constraint is the minimum area constraint,

$$X_c(minimum\_area - X_c) \leq 0. \tag{7}$$

This constraint limits the crop planted to be of at least a minimum size or zero. This means that if a crop has a minimum plantable area, the corresponding crop area, $X_c$, must be greater than this minimum area if the crop is to be planted.

The following constraint relates to the amount of groundwater pumping. The pumped groundwater needed can be obtained from the accessible surface water and the crop water requirements for irrigation of the crop in a month (m) and is given by

$$P_m = \left( \sum_{c=1}^{C} WREQ_{c,m} X_c \right) - Allocation(m). \tag{8}$$

The last constraint is the water allocation constraint,

$$Allocation(m) = Inflow(m) - Env\_f(m). \tag{9}$$

After the environmental flow is released from the accessible surface river water, the remaining water can be used to irrigate the crops in a month (m).

Using Eqs (4) and (9), we write (8) as

$$P_m = (\sum_{c=1}^{C} (k_{c,m} ET_m - Rain_m)X_c) - (Inflow(m) - Env\_flow\_f(m)).$$

Considering Eqs (2) and (3) which are the objectives of our model and combine with (8) and (9), subject to the constraints Eqs (5)—(7), we formulate the bi-objective Problem ($P_{c,m}$) described as follows.

$$min\ [f_1, f_2]$$

$$f_1 = - \sum_{c=1}^{C} TCI_c X_c + C_p \sum_{m=1}^{M} P_m + \sum_{c=1}^{C} Vcost_c X_c$$

$$+ C_w(Inflow(m) - Env\_flow\_f(m))$$

$$f_2 = \sum_{m=1}^{M} max[(Ter\_env\_flow\_f(m) - Env\_flow\_f(m)), 0]$$

Subject to

$$\sum_{m=1}^{M} P_m \leq total\_Pump$$

**Table 1. Rainfall data (in mm) in the Muhuri irrigation area.**

| Jan | Feb | Mar | Apr | May | Jun | Jul | Aug | Sep | Oct | Nov | Dec |
|-----|-----|-----|-----|-----|-----|-----|-----|-----|-----|-----|-----|
| 0 | 28.27 | 19.5 | 298.25 | 313.5 | 508.75 | 887.25 | 442.75 | 340.75 | 370.75 | 5.75 | 56.25 |

$$\sum_{c=1}^{C} X_c \leq T_{Area}$$

$$X_c(minimum\_area - X_c) \leq 0$$

$$X_c, \; Env\_flow\_f(m) \geq 0$$

## 5. Model solution and experimental setup

The annual average rainfall in the MIP area is 2447 mm [20]. Here we use average rainfall data collected from the Bangladesh Water Development Board (BWDB), Feni, Bangladesh, as provided in Table 1.

Evapotranspiration is the sum of the water evaporated from the soil and plant and transpired through the plant. Evapotranspiration reaches the maximum level in April and May when temperature, sunshine, and wind are at or close to their maximum levels for the year. Monthly evapotranspiration data was collected from [5] and provided in Table 2.

Major rivers within the project area are the Feni, Kalidas-Pahalia, and Muhuri rivers.

In addition, there are many Khals located in the area. Other rivers outside the project area, such as Titas, Gumti, Dakatia and Meghna, act as the prominent drainage collectors. Surface water irrigation is from the three rivers and supported by storage in the rivers, drains and reservoirs in the backwater from Feni Regulator. Table 3 contains the water inflows from the three rivers which were collected from [5].

The crop coefficient, $k_{c,m}$, is the ratio of the reference crop evapotranspiration, $ET_0$ and crop evapotranspiration, $ET_c$.

$$k_{c,m} = \frac{ET_c}{ET_0}$$

In this research, crop coefficient data in Table 4 has been taken from [20].

Crops production (T/ha) and crop market price (AUD) data in Table 5 were collected from the Deputy Chief Extension Officer, BWDB, Feni, Bangladesh.

The number of variables set in this study is the total number of crops, $X_c$ which consists of ten crops and the environmental flow, $Env\_flow\_f(m)$, for twelve months. The lower bound of all the variables is zero. The upper bound of the cultivable area for each crop is 70,000 ha. The minimum area is 1000 ha. The target environmental flow, $Ter\_env\_flow\_f(m)$ is set to 100 GL for each month.

The Problem $(P_{c,m})$ is a multi-objective nonlinear constrained optimisation problem, requiring an excellent computational method to approximate the Pareto solutions. This article uses the NSGA-II for solving the Problem $(P_{c,m})$. Deb et al. [21] developed the NSGA-II, a

**Table 2. Evapotranspiration (in mm) data in the Muhuri irrigation area.**

| Area | Jan | Feb | Mar | Apr | May | Jun | Jul | Aug | Sep | Oct | Nov | Dec |
|------|-----|-----|-----|-----|-----|-----|-----|-----|-----|-----|-----|-----|
| Feni | 72 | 89 | 130 | 143 | 145 | 115 | 113 | 117 | 110 | 106 | 81 | 68 |

**Table 3. Total water inflow in cubic meters.**

| Jan | Feb | Mar | Apr | May | Jun | Jul | Aug | Sep | Oct | Nov | Dec |
|------|------|------|------|------|------|------|-------|------|------|------|------|
| 16.5 | 11.4 | 10.0 | 14.1 | 21.1 | 58.3 | 68.8 | 105.2 | 61.9 | 50.4 | 30.5 | 20.1 |

multi-objective genetic algorithm for solving optimisation problems. The NSGA-II works by dominance and non-dominance relation and to determine Pareto solutions. It is an extension and improvement of NSGA, proposed earlier by Srinivas and Deb [22]. Also, it is an elite and fast sorting multi-objective genetic algorithm. The NSGA-II has three unique properties: simple crowded comparison operator, fast non-dominated sorting approach, and fast crowded distance estimation procedure [21]. The pseudocode of the NSGA-II is given next.

Step1: Randomly create an initial population $P_0$ of size $N$

Step2: Calculate the values of the objective of each individual $P_0$

Step3: By using a non-domination sorting process, assign a rank of each individual $P_0$

Step4: Generate child population $Q_t$ of size $N$ using crossover and mutation

Step5: Calculate the objective values of each child population $Q_t$

Step6: Combine the initial and child population ($P_t = (P_0 \cup Q_t)$) of size $2N$

Step7: Assign rank to each individual $P_t$ based on the non-domination sorting process

Step8: Calculate the crowded distance of individuals in each front

Step9: Select the best $N$ individuals base on rank and crowded distance

Step10: Repeat Step2 to Step9 until the stopping criterion met

Step11: Terminate the algorithm

The population size is a sensitive issue in the genetic algorithm (GA); smaller populations result in lower accuracy of the solution; this means little search space is available. Therefore, it is possible to reach an unwanted local optimum. The further increase in the population size increases the accuracy of the solution, but the computational load becomes high [23]. Therefore, the size of the population must be reasonable. In each computation run, the population size of the algorithm in this study is set at 100.

**Table 4. Crop coefficient $k_{c,m}$ [20].**

| Crops | Jan | Feb | Mar | Apr | May | Jun | Jul | Aug | Sep | Oct | Nov | Dec |
|-------|------|------|-----|-----|-----|-----|------|------|-----|-----|------|------|
| T. Aus | 0.2 | 0.2 | 0.2 | 1.05 | 1.2 | 1.2 | 0.9 | 0.2 | 0.2 | 0.2 | 0.2 | 0.2 |
| T. Aman | 0.2 | 0.2 | 0.2 | 0.2 | 0.2 | 0.2 | 1.2 | 1.2 | 0.9 | 0.2 | 0.2 | 0.2 |
| Boro Rice | 1.2 | 0.9 | 0.9 | 0.2 | 0.2 | 0.2 | 0.2 | 0.2 | 0.2 | 0.2 | 1.05 | 1.2 |
| Wheat | 1.15 | 1.15 | 0.4 | 0.4 | 0.2 | 0.2 | 0.2 | 0.2 | 0.2 | 0.2 | 0.2 | 0.4 |
| Potato | 0.8 | 0.8 | 0.2 | 0.2 | 0.2 | 0.2 | 0.2 | 0.2 | 0.2 | 0.6 | 1.15 | 1.15 |
| Oilseeds | 0.3 | 0.2 | 0.2 | 0.2 | 0.2 | 0.2 | 0.2 | 0.2 | 0.2 | 0.2 | 0.35 | 1.05 |
| Pulses | 1.05 | 0.3 | 0.3 | 0.2 | 0.2 | 0.2 | 0.2 | 0.2 | 0.2 | 0.2 | 0.2 | 0.4 |
| Sugarcane | 0.4 | 0.4 | 0.2 | 0.2 | 0.2 | 0.2 | 1.15 | 1.15 | 0.9 | 0.9 | 0.6 | 0.6 |
| Winter Vegetable | 0.9 | 0.9 | 0.2 | 0.2 | 0.2 | 0.2 | 0.2 | 0.2 | 0.2 | 0.2 | 0.6 | 1.1 |
| Summer Vegetable | 0.2 | 0.2 | 0.2 | 0.6 | 1.1 | 0.9 | 0.9 | 0.2 | 0.2 | 0.2 | 0.2 | 0.2 |

**Table 5. Economic data for crops in the Muhuri irrigation area (1 AUD = 60 Taka).**

| Crops | Production (T/Ha) | Market Price (AUD) |
|---|---|---|
| (1) T. Aus | 3.2 | 331 |
| (2) T. Aman | 4.25 | 365 |
| (3) Boro Rice | 5.85 | 331 |
| (4) Wheat | 2.8 | 206 |
| (5) Potato | 23 | 248 |
| (6) Oilseeds | 1.1 | 537 |
| (7) Pulses | 1.56 | 557 |
| (8) Sugarcane | 50 | 4965 |
| (9) Winter Vegetable | 16.5 | 435 |
| (10) Summer Vegetable | 14.85 | 383 |

The crossover rate (probability) is a genetic operator used to vary the programming of a chromosome or chromosomes from one generation to the next, i.e., the chance that two chromosomes exchange some parts if crossover probability is 100%, then all offspring are made by crossover. If it is 0%, a whole new generation is made from exact copies of chromosomes from the old population, except those that resulted from the mutation process. The crossover rate is in the range of [0, 1] [24]. The crossover rate in this study is set at 0:2.

The mutation is another vital operator which takes place after the crossover is done. The mutation rate decides how many chromosomes should be mutated in one generation. The mutation rate is in the range of [0, 1] [25]. In our study, the mutation scaling factor is set at 1.

The number of generations refers to the number of cycles before the algorithm stops. It depends on the type of optimisation problem and its complexity. In this case, the NSGA-II algorithm is iterated for 500 generations. It is to note here that setting the frequency of change based on the number of generations sometimes makes the comparison unfair. However, our experience shows that the more the population size and the number of generations, the more the results converge. Therefore we use the number of generations instead of function evaluations.

For evolutionary algorithms like GA, there are seven kinds of stopping criteria [26]. In this research, the maximum number of iterations is set for stopping criteria, and it is 300 iterations.

## 6. Results and discussion

In Section 4, we have demonstrated the multi-objective optimisation problem ($P_{c,m}$) for the Muhuri Irrigation Project (MIP). Our objectives have been maximising net return (NR) and minimising deficit in environmental flow (EFD) under constraints. We have adopted the NSGA-II algorithm for solving the Problem ($P_{c,m}$). Our experimental results are as follows:

### Results

The test run was carried out using 300 iterations. The Pareto front obtained for 300 iterations is demonstrated in Fig 1, and we have considered this Pareto front as a base level solution. The information on the number of solutions, the computational time, and the range of objective function values obtained are in Table 6 for the NSGA-II algorithm. The Pareto front is taken from NSGA-II, representing 34 non-dominated solutions for net return in units of 10 million Australian dollars and environmental flow deficit in units of 100 GL.

Table 6 shows that when the maximum net return is $1877.48 \times 10^7$ AUD, the environmental flow deficit increases to a maximum of 35.53 GL. In such a case, one needs to compromise

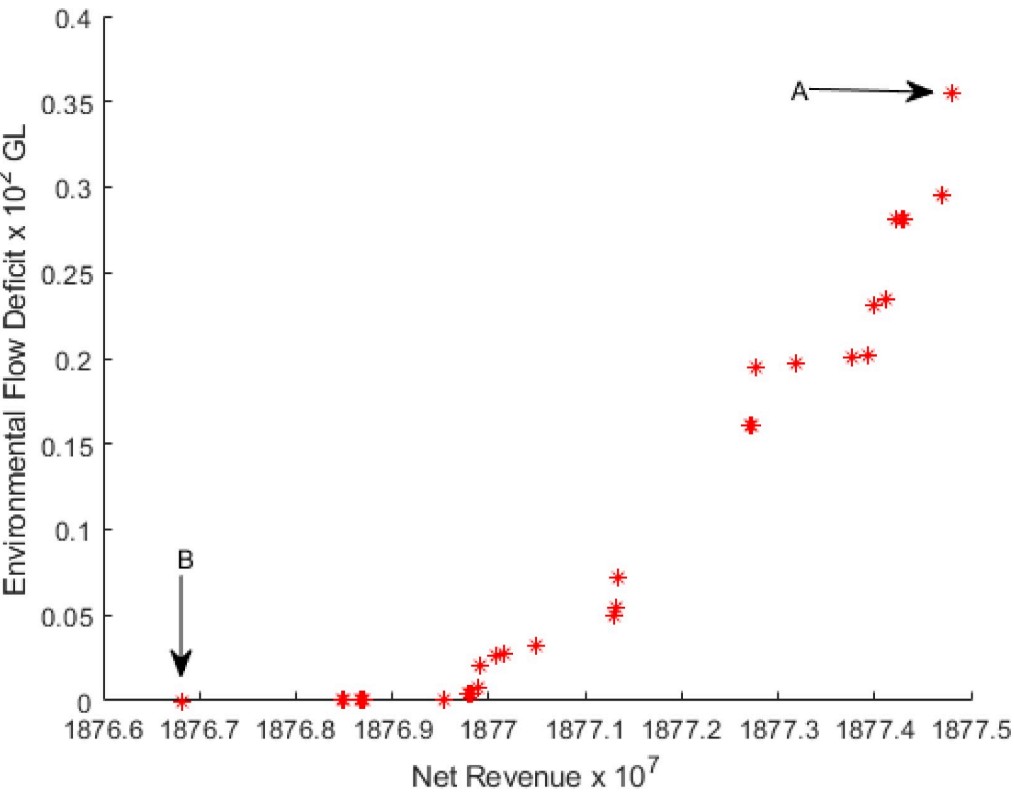

**Fig 1. Pareto front for 300 iterations.**

with the environmental flow. On the other hand, we can keep EFD on it lower, in which case, the net return would be $1876.68 \times 10^7$ AUD, which is the lowest net return on the Pareto front.

The solution of the MOPs is a set of efficient solutions, which are also Pareto optimal solutions. There is a role of a decision-maker in choosing a solution among many options. We cannot say one solution is better than the other in this experiment. Only the decision-maker identify the best solution depends on their preference.

According to Fig 1, the analysis of all the 34 solutions, solution 1 (A in Fig 1) shows the best in terms of net return (NR) but worst in terms of environmental flow deficit (EFD). Whilst solution 34 (B in Fig 1) is the best in EFD but worst in NR.

**Crop area.** Cropping patterns are used for the MIP to approximate the Pareto front shown in Fig 1. The 1st solution (A in Fig 1) of the S1 Table included in the Supporting information file suggests that T. Aus, T. Aman, Boro Rice, Wheat, Potato, Oilseeds, Pulses, Sugarcane, Winter Vegetables, and Summer Vegetables should be planted in 1452.18 (ha), 1516.63 (ha), 13504.46 (ha), 2555.28 (ha), 48610.52 (ha), 6567.29 (ha), 1072.37 (ha), 69228.00 (ha),

**Table 6. Summary for 300 iterations.**

| Objectives | Net return (NR) | Environmental flow deficit (EFD) |
|---|---|---|
| Mean | $1877.12 \times 10^7$ AUD | 10.01 GL |
| Maximum output | $1877.48 \times 10^7$ AUD | 35.53 GL |
| Minimum output | $1876.68 \times 10^7$ AUD | 0.00 GL |
| Number of solutions | 34 | |
| Computational time | 706.59 minutes | |

69227.79 (ha) and 16982.25 (ha) areas of land respectively. When we inspect the solution for the crop mix, we see that the maximum areas, 69228.00 (ha) and 69227.79 (ha), are devoted to growing Sugarcane and Winter Vegetables. The reason becomes clear as both crops are highly profitable and the production is a high per hectare of 50 tonnes and 16.5 tonnes respectively. Also, Sugarcane and Winter Vegetables provide a gross return of AUD 4965 and AUD 435 per hectare.

The 34th solution (B in Fig 1) has the lowest net return of AUD $1876.68 \times 10^7$ with zero GL deficit in environmental flow. The cropping pattern of the 34th solution as provided in the S2 Table included in the Supporting information file, suggests that T. Aus, T. Aman, Boro Rice, Wheat, Potato, Oilseeds, Pulses, Sugarcane, Winter Vegetables, and Summer Vegetables should be planted in 1453.53 (ha), 1529.59 (ha), 13451.38, (ha), 2854.91 (ha), 48197.63 (ha), 6631.99 (ha), 1072.15 (ha), 69227.99 (ha), 69227.88 (ha) and 17026.76 (ha) areas of land respectively. A few differences between these two solutions are noticeable. The 1st solution (A) presents the planting area of Wheat, Potato, and Summer Vegetables at 2555.28 (ha), 48610.52 (ha), and 16982.25 (ha), respectively. However, in the 34th solution (B), a slightly different scenario is seen for planting these three crops. Here 2854.91 (ha), 48197.63 (ha), and 17026.76 (ha) areas of land are devoted to these crops.

**Environmental flow.**　The environmental flow for the Pareto front of Fig 1 is provided in the Supporting Information file. The 1st solution (A in Fig 1) of the S3 Table in the Supporting Information file shows that the highest amount of water, i.e. approximately 250 GL, is required for environmental flow in November. The second and third most elevated amount of water is needed for June and October, and their amount is approximately 164 GL and 145 GL, respectively. About 129 GL water is required for the month of May. Finally, the environmental flow is almost the same (near 100 GL) in the remaining months.

We see a slight difference of environmental flow in GL of the 34th solution (B in Fig 1) is given in the S4 Table. In November, approximately 256 GL of water is needed for the environmental flow, which is the highest amount of water across all other months. About 162 GL and 157 GL are required for June and October, respectively. About 145 GL of water is needed for May. Finally, the environmental flow is almost the same for approximately 100 GL for the rest of the year.

## Effect of rainfall

The results for five Pareto front curves when rainfall is varied by 10% and 20% above and below the base level using 300 simulation run is shown in Fig 2.

Fig 2 illustrates that if rainfall is 10% and 20% below the base level, then for the 1st solution, NR will decrease 0.47% and 0.76%, respectively, whereas EFD will increase 67.95% and 71.02%, respectively.

Also, if rainfall is 10% and 20% above the base level, NR will increase 0.54% and 0.77%, respectively. On the other hand, EFD will decrease by 37.63% and 28.27%.

The crop pattern for the 1st solution in the net return for different rainfall using 300 simulation runs is provided in Fig 3A. According to Fig 3A, the land area for cultivating Sugarcane is the same for all five conditions at approximately 69228 ha. However, the most significant difference is observed for Potatoes and Summer Vegetables. In the base level rainfall, we see the highest amount of land is devoted to cultivating crop 5 (Potatoes), but the opposite scenario is seen for Summer Vegetables. When rainfall decreases or increases, the cultivation of Potatoes continuously decreases, but the opposite happens for Sugarcane. For other crops, the differences are minor but still varied.

The environmental flow for the 1st solutions in the context of the net return for different rainfall using 300 simulation runs are provided in Fig 3B. As observed from Fig 3B, when rain

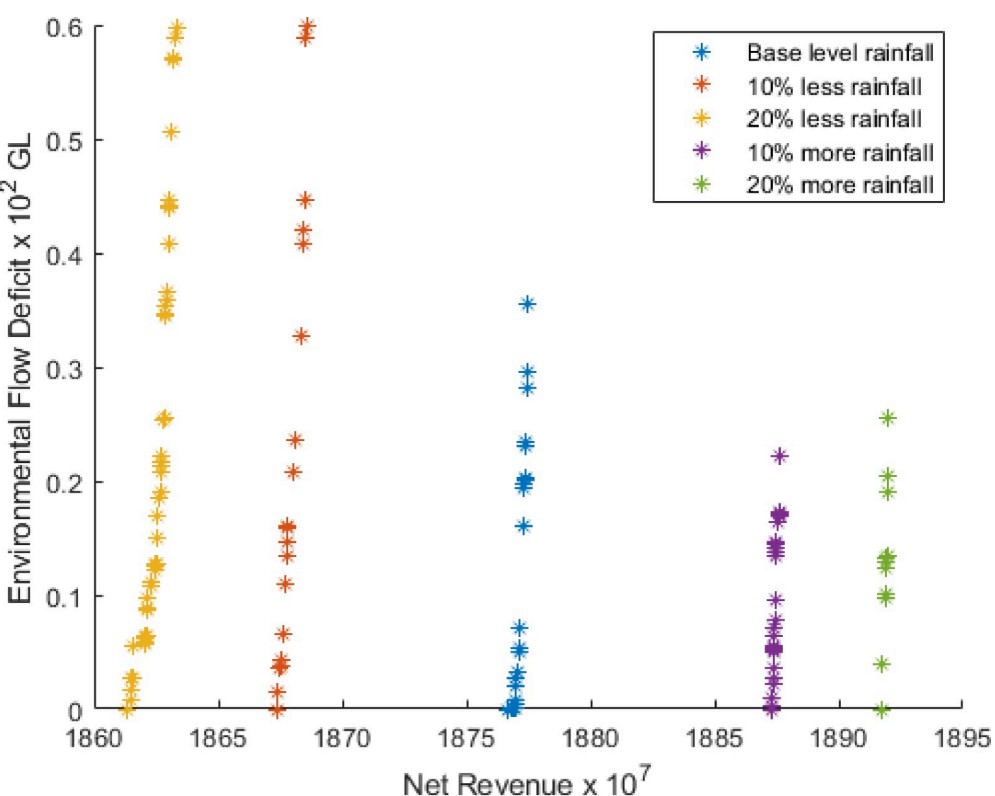

**Fig 2. Pareto fronts for different rainfall.**

is 20% above the base level, the highest environmental flow is required for the month 5 (May) at approximately 290 GL. On the other hand, the lowest environmental flow is needed for month 3 (March) when rainfall is 10% below the base level at about 50 GL.

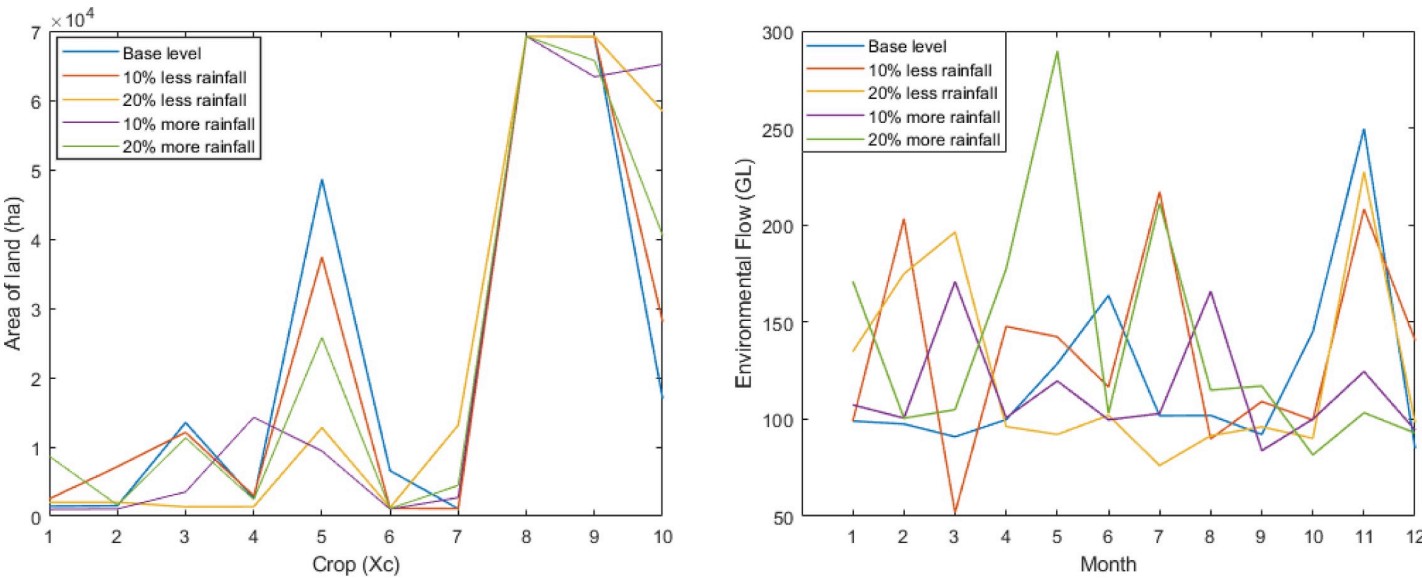

**Fig 3. Effect of different rainfall on crops and environmental flow.** (a) Crops pattern for different rainfall. (b) Environmental flow for different rainfall.

In light of the above discussion, it can be argued that if it rains more, profits will increase, and the cost of irrigation and water supply for environmental flow will decrease.

## Effect of water inflow

The results for five Pareto front curves when water inflow is varied by 10% and 20% above and below the base level using 300 simulation run is shown in Fig 4.

Fig 4 illustrates that if water inflow is 10% and 20% below the base level, NR will decrease 0.21% and 0.37%, and EFD will decrease 11.71% and 60.66%, respectively. In addition, if water inflow is 10% and 20% above the base level, NR will increase 0.36%, and $5.32 \times 10^{-9}$ % and EFD will decrease by 2.81% in both cases.

According to Fig 4, the highest environmental flow is required for less than 10% water inflow from the base level in the month 6 (June). The same scenario is seen for base-level water inflow in month 11 (November). For the case of 10% more water inflow, we see more than 200 GL water is required for environmental flow in months 2 (February) and 12 (December). From the above discussion, we conclude that more water inflow brings more profit.

The crops pattern for the 1st solutions in the context of the net return for different water inflows using 300 simulation runs is provided in Fig 5A. Based on Fig 5A, we see the same scenario with slight differences. For different water inflow level conditions, Sugarcane is cultivated across the same area of land. However, for Potatoes and Summer Vegetables, the opposite occurs. For all other crops, there is a slight variation.

The environmental flow for the 1st solutions in the context of the net return for different water inflow using 300 simulation runs is provided in Fig 5B. According to Fig 5B, the highest

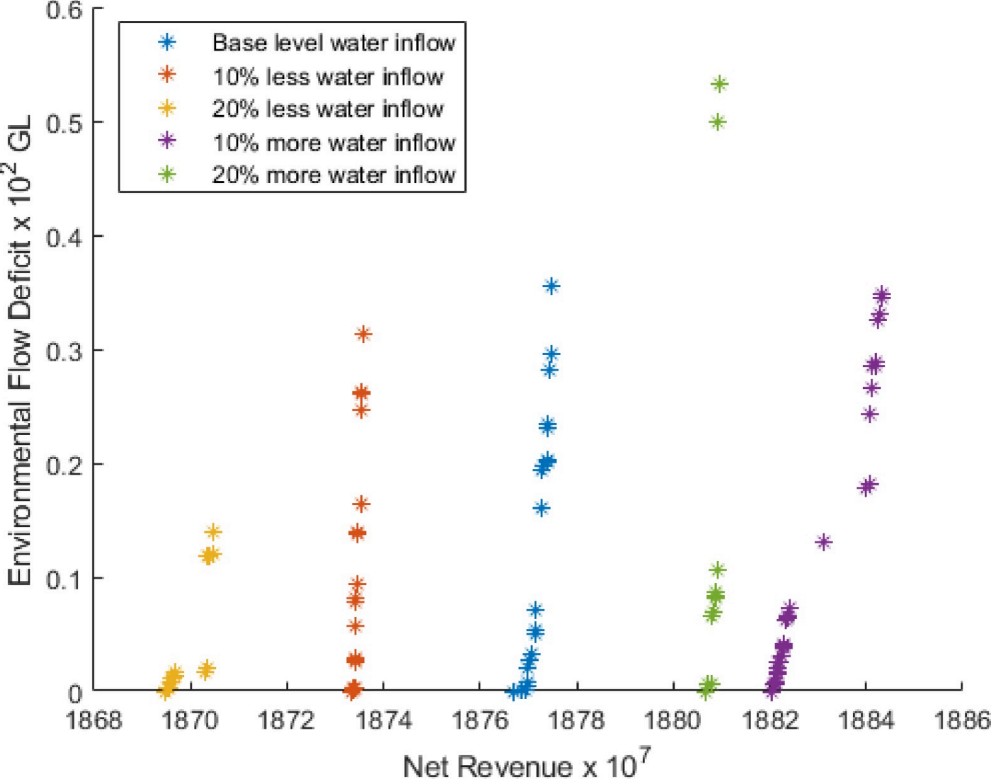

**Fig 4. Pareto front for different water inflow.**

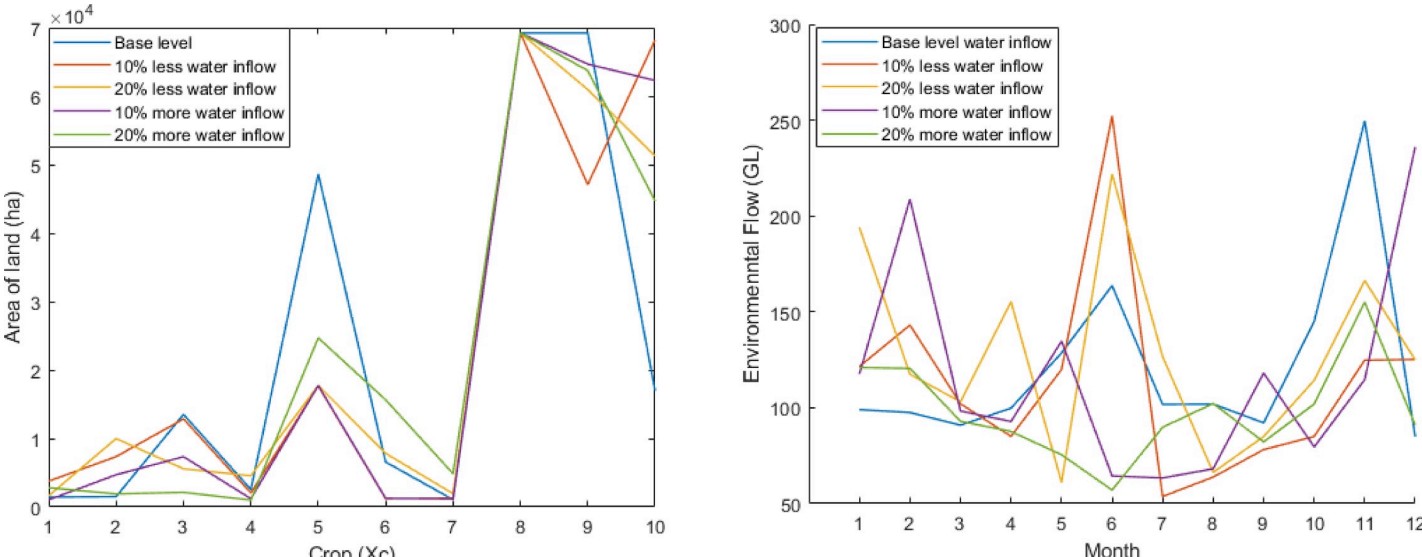

**Fig 5. Effect of different water inflow on crops and environmental flow.** (a) Crops pattern for different water inflow. (b) Environmental flow for different water inflow.

environmental flow is required for less than 10% water inflow from the base level in month 6 (June). The same scenario is seen for base-level water inflow in the month 11 (November). For the case of 10% more water inflow, we see more than 200 GL of water is required for environmental flow in months 2 (February) and 12 (December).

As expected, this leads to the notion that more water inflow brings more profit.

## 7. Conclusion

This study sought to explore the economics of optimal water allocation for irrigation and optimal cropping patterns in the MIP of Bangladesh. Although Bangladesh is not a country with widespread, year-round water scarcity, it faces severe water shortages during the dry winter season. This article aims to maximise net return and minimise the deficit in environmental flow using optimal water management policies.

Based on the framework mentioned above, the research has several outcomes. The following is a synthesis of those outcomes:

- The crop which produces the most significant profitability is recommended to be cultivated to a greater extent

- During the dry season, more environmental flow is required to sustain the environment and to grow crops than in the rainy season.

- The decrease and increase of net return (NR) and rainfall are directly proportional to each other. However, the relationship between rainfall and environmental flow deficit (EFD) is not proportional. The decrease of rain by 10% contributes to the increase of environmental flow deficit (EFD), but the decrease of rain by 20% does not impact the environmental flow deficit (EFD) in the same way.

- When water inflows increase, net returns (NR) also increase. On the other hand, the environmental flow deficit (EFD) decreases with increased water inflow and vice versa.

## Supporting information

**S1 Table. Details of 1–17 Pareto solutions for the crops.** This includes results that we have found from using the Non-dominated Sorting Genetic Algorithm-II (NSGA-II) on the Multi-objective Optimisation Problem (MOP).
(PDF)

**S2 Table. Details of 18–34 Pareto solutions for the crops.** This includes results that we have found from using the Non-dominated Sorting Genetic Algorithm-II (NSGA-II) on the Multi-objective Optimisation Problem (MOP).
(PDF)

**S3 Table. Details of 1–17 Pareto solutions for the environmental flow.** This includes results that we have found from using the Non-dominated Sorting Genetic Algorithm-II (NSGA-II) on the Multi-objective Optimisation Problem (MOP).
(PDF)

**S4 Table. Details of 18–34 Pareto solutions for the environmental flow.** This includes results that we have found from using the Non-dominated Sorting Genetic Algorithm-II (NSGA-II) on the Multi-objective Optimisation Problem (MOP).
(PDF)

## Acknowledgments

The authors are very thankful to Dr Trevor Langlands, Senior Lecturer (Mathematics) from the School of Sciences, University of Southern Queensland, Australia, for his guidance during this research. Also, we would like to express our deep gratitude to Dr Mohammed Mustafa Rizvi, Associate Professor, Department of Mathematics, University of Chittagong, Bangladesh and research fellow from UniSA STEM, University of South Australia, Australia, for his inspiration during this research. Moreover, the authors are grateful for the valuable comments and suggestions from the respected editor and reviewers. Their helpful comments and suggestions have enhanced the strength and significance of this paper.

## Author Contributions

**Conceptualization:** G. M. Wali Ullah.

**Data curation:** G. M. Wali Ullah.

**Formal analysis:** G. M. Wali Ullah.

**Writing – original draft:** G. M. Wali Ullah.

**Writing – review & editing:** Micah Nehring.

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
