## [Decision Letter · Decision Letter 0]

25 Apr 2021

PONE-D-21-08160

A multi-objective mathematical model of a water management problem with environmental impacts: An application in an irrigation project

PLOS ONE

Dear Dr. Ullah,

Thank you for submitting your manuscript to PLOS ONE. After careful consideration, we feel that it has merit but does not fully meet PLOS ONE’s publication criteria as it currently stands. Therefore, we invite you to submit a revised version of the manuscript that addresses the points raised during the review process.

Please revise the work thoroughly according to the reviewers' comments. 

We look forward to receiving your revised manuscript.

Kind regards,

Shouyong Jiang, PhD

Academic Editor

PLOS ONE

Journal Requirements:

PLOS requires an ORCID iD for the corresponding author in Editorial Manager on papers submitted after December 6th, 2016. Please ensure that you have an ORCID iD and that it is validated in Editorial Manager. To do this, go to ‘Update my Information’ (in the upper left-hand corner of the main menu), and click on the Fetch/Validate link next to the ORCID field. This will take you to the ORCID site and allow you to create a new iD or authenticate a pre-existing iD in Editorial Manager. Please see the following video for instructions on linking an ORCID iD to your Editorial Manager account: https://www.youtube.com/watch?v=_xcclfuvtxQ

Additional Editor Comments :

Obviously there are a number of issues which need to be well addressed for publication.

Reviewers' comments:

Reviewer's Responses to Questions

**Comments to the Author**

1. Is the manuscript technically sound, and do the data support the conclusions?

Reviewer #1: Partly

Reviewer #2: Yes

2. Has the statistical analysis been performed appropriately and rigorously? 

Reviewer #1: I Don't Know

Reviewer #2: Yes

3. Have the authors made all data underlying the findings in their manuscript fully available?

Reviewer #1: Yes

Reviewer #2: Yes

4. Is the manuscript presented in an intelligible fashion and written in standard English?

Reviewer #1: Yes

Reviewer #2: Yes

5. Review Comments to the Author

Reviewer #1: This work seems very significant to address optimum water allocation in the Muhuri Irrigation Project (MIP), Bangladesh. However, it still exists several problems.

1. Only seeing the current abstract of this paper, I don,t think that readers can appreciate it since the abstract about this work is very unclear for interested readers. So, I suggest the author improve its abstract to make it more persuasive.

2. Related work needs further discussion. Some important developments

in this area in recent years do not seem to be discussed. Additionally，half of the provided literature during the current manuscript is elder, which suggests that the provided literature of the author haven't authoritative.

3. This paper hasn't provided any parameters illustration. Some common parameters should be mentioned. Please, discuss how the parameters

of this paper were settled, and if this paper is robust to parameters

values change or not.

4. Please guaranty that you explains each symbol in all formulas of this paper. If the problem is not corrected next time, I will be obliged to reject the paper.

5. Authors should add a paragraph into the introduction section. They should write, "The main contributions of this paper are: (i) ….. (ii) ……. and (iii) ……" to highlight the key works. By this way, authors should provide a stronger motivation clearly and explain the originality of the paper.

6. Setting the frequency of change based on the number of generations sometimes makes the comparison unfair. It is preferable to be based on the number of function evaluations, or you should explain why it's still fair to use the number of generations.

Reviewer #2: After reading this paper, it is recommended to consider accepting the following major revisions:

1.Check the English sentences carefully. There are too many words repeated in the sentences in a paragraph. It is recommended to use refined sentences to write. Suggest additional content in the abstract section.

2.In the article, it is recommended to increase the algorithm pseudo code, and can make improvements on the genetic operator.

3.It is found from Table 6 that the "Computational time" time is too long. Can the algorithm be improved to reduce the CPU time?

6. PLOS authors have the option to publish the peer review history of their article (what does this mean?). If published, this will include your full peer review and any attached files.

Reviewer #1: No

Reviewer #2: No

---

## [Author Response · Author response to Decision Letter 0]

27 May 2021

Dear Editor and Reviewers,

We would like to thank you for the careful review and thoughtful feedback on our manuscript “A multi-objective mathematical model of a water management problem with environmental impacts: An application in an irrigation project” (PONE-D-21-08160) and the opportunity to resubmit a revised copy. We have revised the manuscript according to the comments and believe that it is substantially improved by incorporating these edits.

Below, we provide a point-by-point reply to the reviewers’ comments. We have included a marked copy of the revised manuscript that highlights changes and a clean version. We have also ensured that our manuscript meets the style requirements of the PLOS ONE.

Thank you for your consideration of our revised manuscript.

EDITOR COMMENTS

Comment 1: Please ensure that your manuscript meets PLOS ONE's style requirements, including those for file naming. The PLOS ONE style templates can be found at https://journals.plos.org/plosone/s/file?id=wjVg/ POSOne_formatting_sample_main_body.pdf 

and

Response: We have prepared our manuscript as per the PLOS ONE journal requirements. The file name is Manuscript.docx.

Comment 2: PLOS requires an ORCID iD for the corresponding author in Editorial Manager on papers submitted after December 6th, 2016. Please ensure that you have an ORCID iD and that it is validated in Editorial Manager. To do this, go to ‘Update my Information’ (in the upper left-hand corner of the main menu), and click on the Fetch/Validate link next to the ORCID field. This will take you to the ORCID site and allow you to create a new iD or authenticate a pre-existing iD in Editorial Manager. Please see the following video for instructions on linking an ORCID iD to your Editorial Manager account: https://www.youtube.com/watch?v=_xcclfuvtxQ

Response: ORCID iD https://orcid.org/0000-0001-5491-1289

REVIEWER # 1 COMMENTS

Comment 1. Only seeing the current abstract of this paper, I don,t think that readers can appreciate it since the abstract about this work is very unclear for interested readers. So, I suggest the author improve its abstract to make it more persuasive.

Response: We have rewritten the abstract in the following way on page 2 (lines 25 -35) in the Manuscript file:

“The study proposes applying an efficient but straightforward multi-objective constrained optimisation model for optimal water allocation among irrigation and environmental sectors. The model has been implemented in the Muhuri Irrigation Project (MIP), Bangladesh, where the irrigation systems lead to unjustifiable use of groundwater. This study explores how water can be optimised to increase agricultural production and sustain the local environment in the MIP. Hence, the paper has two objectives--to maximise the net return and minimise the deficit in environmental flow. The study uses a Non-Dominating Sorting Genetic Algorithm, NSGA-II, to solve the research problem. Results indicate that crops more profitable to trade should be cultivated. The rainfall has more impact on the net return and environmental flow deficit than water inflow. The findings of this study can help plan irrigation water and cropland resources and be a reference for further studies.”

Comment 2: Related work needs further discussion. Some important developments in this area in recent years do not seem to be discussed. Additionally, half of the provided literature during the current manuscript is elder, which suggests that the provided literature of the author haven't authoritative.

Response: Three more recent articles published in 2021 are included in the reference section on page 25, reference numbers 14 - 16. Also, six lines are included about these articles on page 4 (lines 109 - 114) in Manuscript file as follows:

“Musa [14] applied a multi-objective model in Saudi Arabia for optimal water allocation in three sectors named domestic sector, agriculture sector, and industrial sector. A goal programming technique has been used to solve this problem. Marzban et al. [15] proposed an optimal cropping pattern of irrigation and rainfed crops using multi-objective nonlinear programming to minimise environmental impact and maximise the revenue in Iran.” 

Comment 3: This paper hasn't provided any parameters illustration. Some common parameters should be mentioned. Please, discuss how the parameters of this paper were settled, and if this paper is robust to parameters values change or not.

Response: More information about NSGA-II parameters is given on pages 15 and 16 (lines 318 - 343) in the Manuscript file as follows: 

“The population size is a sensitive issue in the genetic algorithm (GA); the use of smaller populations results in lower accuracy of the solution, this means little search space is available, and therefore it is possible to reach an unwanted local optimum. The further increase in the population size increases the accuracy of the solution, but the computational load becomes high [23]. Therefore, the size of the population must be reasonable. In each computation run, the population size of the algorithm in this study is set at 100.

The crossover rate (probability) is a genetic operator used to vary the programming of a chromosome or chromosomes from one generation to the next, i.e., the chance that two chromosomes exchange some of their parts, If crossover probability is 100%, then all offspring is made by crossover. If it is 0%, whole new generation is made from exact copies of chromosomes from old population, except those resulted from the mutation process. The crossover rate is in the range of [0, 1] [24]. The crossover rate in this study is set at 0:2.

The mutation is another vital operator which takes place after the crossover is done. The mutation rate decides how many chromosomes should be mutated in one generation. The mutation rate is in the range of [0, 1] [25]. In our study, the mutation scaling factor is set at 1. 

The number of generations refers to the number of cycles before the algorithm stops. It depends on the type of optimisation problem and its complexity. In this case, the NSGA-II algorithm is iterated for 500 generations. It is to note here that setting the frequency of change based on the number of generations sometimes makes the comparison unfair. However, our experience shows that the more the population size and the number of generations, the more the results converge. Therefore, we use the number of generations instead of function evaluations. 

For evolutionary algorithms like GA, there are seven kinds of stopping criteria [26]. In this research maximum, the number of iterations is set for stopping criteria, and it is 300 iterations.”

Comment 4: Please guaranty that you explain each symbol in all formulas of this paper. If the problem is not corrected next time, I will be obliged to reject the paper.

Response: We have added a new section named “List of symbols” on pages 7 and 8 (lines 167 - 188) in the Manuscript.docx file.

Comment 5: Authors should add a paragraph into the introduction section. They should write, "The main contributions of this paper are: (i) ….. (ii) ……. and (iii) ……" to highlight the key works. By this way, authors should provide a stronger motivation clearly and explain the originality of the paper.

Response: We have added a paragraph into the introduction section on page 6 (lines 123 -131) in the Manuscript.docx file as follows:

“The main contributions of this paper can be highlighted as follows.

i. The Lewis and Randall [6] model is adopted and improved for this research project and applied in the Muhuri Irrigation Project (MIP), Bangladesh.

ii. Considering the scenarios of different available water resources, the results can have an impact on the agricultural production in the MIP area.

iii. This method is very systematic and applied to different scopes, including water resources management. The most important thing is that the model can be used in other irrigation projects only by modifying the parameters according to the actual situation.”

Comment 6: Setting the frequency of change based on the number of generations sometimes makes the comparison unfair. It is preferable to be based on the number of function evaluations, or you should explain why it's still fair to use the number of generations.

Response: More information is given on page 16 (lines 25 - 35) in the Manuscript.docx file as follows: “It is to note here that setting the frequency of change based on the number of generations sometimes makes the comparison unfair. However, our experience shows that the more the population size and the number of generations, the more the results converge. Therefore, we use the number of generations instead of function evaluations.”

REVIEWER # 2 COMMENTS

Comment 1: Check the English sentences carefully. There are too many words repeated in the sentences in a paragraph. It is recommended to use refined sentences to write. Suggest additional content in the abstract section.

Response: We have checked the English sentences carefully. Also, we have rewritten the abstract in the following way on page 2 (lines 25 -35) in the Manuscript file:

“The study proposes applying an efficient but straightforward multi-objective constrained optimisation model for optimal water allocation among irrigation and environmental sectors. The model has been implemented in the Muhuri Irrigation Project (MIP), Bangladesh, where the irrigation systems lead to unjustifiable use of groundwater. This study explores how water can be optimised to increase agricultural production and sustain the local environment in the MIP. Hence, the paper has two objectives--to maximise the net return and minimise the deficit in environmental flow. The study uses a Non-Dominating Sorting Genetic Algorithm, NSGA-II, to solve the research problem. Results indicate that crops more profitable to trade should be cultivated. The rainfall has more impact on the net return and environmental flow deficit than water inflow. The findings of this study can help plan irrigation water and cropland resources and be a reference for further studies.”

Comment 2: In the article, it is recommended to increase the algorithm pseudo code, and can make improvements on the genetic operator.

Response: The pseudocode of the NSGA-II is added on pages 15 (line number 306 - 317) in the Manuscript file.

Comment 3: It is found from Table 6 that the "Computational time" time is too long. Can the algorithm be improved to reduce the CPU time?

Response: In this research total number of variables is 22, the population size is 100, and the NSGA-II algorithm is iterated for 500 generations. The maximum number of iterations was set for stopping criteria, and it was 300. When we decreased the population size, the number of generation and the total number of iteration, “Computational time” reduced. Then, we ran this program for 600 and 1000 simulations. The computational times were 1324.17 min and 2610.44 min, respectively. A simulation run of the algorithm for 1500 iterations was attempted. Unfortunately, the algorithm failed complete processing due to insufficient memory. The simulations in this research were conducted on a Windows 10 laptop with 8 GB RAM running a 1.60 GHz Intel(R) Core (TM) i5-8250U CPU.

---

## [Decision Letter · Decision Letter 1]

30 Jun 2021

PONE-D-21-08160R1

A multi-objective mathematical model of a water management problem with environmental impacts: An application in an irrigation project

PLOS ONE

Dear Dr. Ullah,

Thank you for submitting your manuscript to PLOS ONE. After careful consideration, we feel that it has merit but does not fully meet PLOS ONE’s publication criteria as it currently stands. Therefore, we invite you to submit a revised version of the manuscript that addresses the points raised during the review process.

We look forward to receiving your revised manuscript.

Kind regards,

Shouyong Jiang, PhD

Academic Editor

PLOS ONE

Journal Requirements:

Additional Editor Comments:

That paper has been largely improved after revision. However, the references need to be consistent. The authors are encouraged to use a referencing style in line with the Journal.

Reviewers' comments:

Reviewer's Responses to Questions

**Comments to the Author**

1. If the authors have adequately addressed your comments raised in a previous round of review and you feel that this manuscript is now acceptable for publication, you may indicate that here to bypass the “Comments to the Author” section, enter your conflict of interest statement in the “Confidential to Editor” section, and submit your "Accept" recommendation.

Reviewer #1: All comments have been addressed

Reviewer #2: All comments have been addressed

2. Is the manuscript technically sound, and do the data support the conclusions?

Reviewer #1: Yes

Reviewer #2: Yes

3. Has the statistical analysis been performed appropriately and rigorously? 

Reviewer #1: Yes

Reviewer #2: Yes

4. Have the authors made all data underlying the findings in their manuscript fully available?

Reviewer #1: Yes

Reviewer #2: Yes

5. Is the manuscript presented in an intelligible fashion and written in standard English?

Reviewer #1: Yes

Reviewer #2: Yes

6. Review Comments to the Author

Reviewer #1: Although the authors already received many comments to improve the readability and presentation of the paper, the literature still needs to be normalized.

Reviewer #2: (No Response)

7. PLOS authors have the option to publish the peer review history of their article (what does this mean?). If published, this will include your full peer review and any attached files.

Reviewer #1: No

Reviewer #2: No

---

## [Author Response · Author response to Decision Letter 1]

12 Jul 2021

Dear Editor and Reviewers,

We would like to thank you for the careful review and thoughtful feedback on our manuscript “A multi-objective mathematical model of a water management problem with environmental impacts: An application in an irrigation project” (PONE-D-21-08160) and the opportunity to resubmit a revised copy. We have revised the manuscript according to the comments and believe that it is substantially improved by incorporating these edits.

Below, we provide a point-by-point reply to the reviewers’ comments. We have included a marked copy of the revised manuscript that highlights changes and a clean version. 

We appreciate your consideration of our revised manuscript.

Journal Requirements:

Response: We have reviewed our reference list as per the PLOS ONE journal requirements in the “Vancouver” style.

Additional Editor Comments:

That paper has been largely improved after revision. However, the references need to be consistent. The authors are encouraged to use a referencing style in line with the Journal.

Response: We have reviewed our reference list as per the PLOS ONE journal requirements. Also, we have tried to be consistent in the references.

REVIEWER COMMENT :

6. Review Comments to the Author:

Reviewer #1: Although the authors already received many comments to improve the readability and presentation of the paper, the literature still needs to be normalized.

Response: We have checked the English sentences carefully and tried to improve the literature.

---

## [Editor Report · Decision Letter 2]

19 Jul 2021

A multi-objective mathematical model of a water management problem with environmental impacts: An application in an irrigation project

PONE-D-21-08160R2

Dear Dr. Ullah,

We’re pleased to inform you that your manuscript has been judged scientifically suitable for publication and will be formally accepted for publication once it meets all outstanding technical requirements.

Kind regards,

Shouyong Jiang, PhD

Academic Editor

PLOS ONE

Additional Editor Comments (optional):

The authors have addressed the issue carefully.
---

## [Editor Report · Acceptance letter]

23 Jul 2021

PONE-D-21-08160R2 

A multi-objective mathematical model of a water management problem with environmental impacts: An application in an irrigation project 

Dear Dr. Ullah:

I'm pleased to inform you that your manuscript has been deemed suitable for publication in PLOS ONE. Congratulations! Your manuscript is now with our production department. 

Kind regards, 

on behalf of

Dr. Shouyong Jiang 

Academic Editor

PLOS ONE